# Genetic characterization and selection of litter size traits of Guizhou Black goat and Meigu goat

Yanpin Zhao[1], Yong Han[2,3]*, Yang Yang[2], Chao Yuan[2], Yong Long[2], Wen Xiao[1]

1 College of Animal Science, Guizhou University, Guiyang, China, 2 Institute of Animal Husbandry and Veterinary Sciences, Guizhou Academy of Agricultural Sciences, Guiyang, China, 3 Guizhou University of Traditional Chinese Medicine, Guiyang, China

* 2642900225@qq.com

**Data Availability Statement:** All relevant data are within the manuscript and its Supporting Information files.

## Abstract

The aim of this study is to explore the genetic characteristics of Guizhou Black goats and Meigu goats and their relationship to reproductive performance through population structure analysis, genetic diversity assessment, and selection signal analysis. Blood samples of 19 Guizhou Black goats and 11 Meigu goats were collected for whole-genome high-throughput sequencing. Using PCA and ADMIXTURE analyses, their population structure and genetic relationships were revealed. Further genetic diversity analysis showed that although there is significant population differentiation, the levels of genetic diversity are similar. Subsequently, these goats were categorized into high-yield and low-yield groups based on their litter sizes, with 15 goats in each group. Then, a selection signal analysis was performed using $F_{ST}$ and π ratios for 33,563 SNP loci. The results identified six candidate genes, including KCNIP4, GFRA2, and DGKH, which are significantly associated with high litter performance. These findings enhanced our understanding of the genetic characteristics and population structure of Guizhou Black goats and Meigu goats. Moreover, they provide an important theoretical foundation and scientific basis for further breeding improvements.

## Introduction

Goats are vital livestock that play a significant role in human life, providing essential resources such as milk, meat, and skins [1]. To meet the growing consumer demand for goat products, the development of superior goat breeds is crucial. Some indigenous goat breeds, despite their excellent meat quality and resistance to local diseases, suffer from persistently low reproductive rates [2]. Although crossbreeding with high-fertility foreign breeds can enhance reproduction, excessive crossbreeding can lead to genome invasion, posing significant risks to the conservation of indigenous breeds [3]. Additionally, traditional selective breeding methods are inefficient and incapable of achieving effective improvements in a short term. Therefore, the application of high-throughput sequencing technology is essential for molecular breeding, aiming to efficiently enhance the reproductive performance of superior indigenous breeds.

**Funding:** Sources of funding for this study:1. Innovative Utilization of Livestock and Poultry Germplasm Resources and Construction of Basic Science and Technology Platform (Qiannongke Technology Innovation [2023] 04). 2.Major Special Projects of the Guizhou Province Department of Science and Technology provided funding for this work (Qianke Service Enterprise [2020] 4009). The funders had no role in study design, data collection and analysis, decision to publish, or preparation of the manuscript.

**Competing interests:** The authors have declared that no competing interests exist.

Litter size is the most direct indicator of reproductive performance. However, this trait is influenced by multiple factors, including ovarian development, ovulation rate, and fertilization rate [4]. These reproductive characteristics are regulated by gonadotropins, ovarian steroid hormones, and growth factors, such as luteinizing hormone (LH) and follicle-stimulating hormone (FSH) [5]. In addition to hormonal effects, mutations in certain genes also impact reproductive traits. For instance, a missense mutation was identified in exon 6 of the POU1F1 gene in SBWC goats. Further analysis revealed that TT genotype does not significantly differ from TG genotype in terms of litter size [6]. A genome-wide SNP screening of Shaanbei White Cashmere goats with varying litter size identified candidate genes including SMAD5, RORA, SOX2, PLD6, IGF2BP1, and RXFP2, which may be involved in regulating reproductive traits [7]. Additionally, a study of Da Zu Black goats using whole-genome sequencing identified 96 candidate genes associated with litter size, such as NR6A1, STK3, and IGF2BP2 [8]. Therefore, whole-genome sequencing can be a powerful tool for identifying markers related to litter size in Guizhou Black goats and Meigu goats. This genetic information will be valuable for molecular breeding and will offer insights into how genetic variations regulate litter size.

The Guizhou black goat is an indigenous black goat breed in the southwest region of Guizhou Province, China. This meat goat breed is characterized by moderate body size, tender meat, adaptability to cold weather and roughage. However, the low fecundity restricts its developmental potential. The Meigu goat, native to Meigu County in Sichuan Province, is known for its large body size, superior meat quality, adaptability and notably higher fertility. Both Guizhou black goat and Meigu goat are black goat types primarily raised for meat production in high-altitude, subtropical monsoon climate of Southwest China. Despite their similarities in appearance, such as hair and horn characteristics, they differ significantly in fertility, with Meigu goats exhibiting better reproductive performance. Therefore, it is crucial to analyze the genetic diversity and selection characteristics related to reproduction in Guizhou black goat and Meigu goat. To date, no studies have explored the genetic diversity and selection characteristics related to litter size in these breeds. In this study, we selected 11 Meigu goats from Gujiao Town, Longli County, Guizhou Province, and 19 Guizhou Black goats from Maiping Town, Huaxi District, Guizhou Province. Based on two years of continuous reproduction records, the goats were divided into high- litter size (HL, n = 15) and low- litter size (LL, n = 15) groups. We performed sequencing using next-generation high-throughput technology, followed by population structure and genetic diversity analyses, and identified candidate genes related to litter size through selection signal analysis. This research not only enhances our understanding and conservation of indigenous goat genetic resources but also provides a scientific basis for selective breeding, thereby improving goat reproductive performance.

## Materials and methods

### Sample collection and full test

Blood samples were collected from 30 black goats, comprising 11 Meigu goats (MG) from Gujiao Town, Longli County, Guizhou Province, and 19 Guizhou Black goats (GZH) from Maiping Town, Huaxi, Guizhou Province, using a vacuum tube containing EDTA and a blood collection needle. Blood collection was performed in an ethical manner and in compliance with animal health and welfare guidelines. Prior approval was obtained from the Experimental Animal Ethics Committee of Guizhou University. The goats collected were healthy 2–3 year old ewes that were regularly tested for disease annually. Although two groups of goats were collected, the feeding and management conditions were the same. litter records from the past two years were obtained, and based on these records, the 30 black goats were categorized into high-yield (HL, n = 15) and low-yield (LL, n = 15) groups in preparation for selective trait

detection. DNA was extracted from the blood samples using the magnetic bead method, and the quality of the DNA samples was assessed through agarose gel electrophoresis and enzyme-labeled detection. Upon passing quality control, the DNA samples were fragmented, end-repaired, and A-tailed, followed by the addition of adapters specific to the DNBseq sequencing platform. The DNA libraries were then enriched through PCR. Finally, the DNA library was sequenced using the DNBSEQ technology (based on DNA nanospheres) developed by BGI and the sequencer DNBSEQ-T7, obtaining an average depth of coverage of 33.3X.

## Genome-wide variant detection and annotation

To minimize errors introduced by human factors during sequencing, we first filtered the raw data using Fastp (v0.23.4) by performing the following steps [9]: (1) removing reads with adapters; (2) discarding paired reads if the N content in the read exceeded 1% of the total base count; (3) discarding paired reads if more than 50% of the bases in the read were of low quality ($Q \leq 5$). Subsequently, the clean reads were aligned to the reference genome (GCF_001704415.2_ARS1.2_genomic.chr) using BWA (v0.7.17), and sequencing depth and genome coverage were assessed with SAMtools (v1.17) to prepare for variant detection [10]. For variant detection, we utilized the HaplotypeCaller module of GATK (v4.2.1.0) to perform multi-sample SNP detection on the processed alignment files [11]. To further reduce analytical errors, we conducted genotype data quality control using Plink (v1.9), retaining only high-quality loci (detection rate $\geq$ 90%, minor allele frequency $\geq$ 0.05) for subsequent analysis. This filtering process resulted in the identification of 13,275,130 SNPs in Meigu goats and Guizhou Black goats. Finally, the filtered SNPs were annotated using ANNOVAR [12].

## Population structure and genetic diversity

Principal component analysis (PCA) was conducted using GCTA (v1.94), with results visualized in R [13]. Genetic distance matrices based on Identity-by-State (IBS) were generated using Plink (v1.9). These matrices were utilized to construct a Neighbor-Joining (NJ) phylogenetic tree, which was visualized with the online tool iTOL (available at: https://itol.embl.de/). Population structure analysis was performed using the Admixture software to infer the ancestral components of individuals based on maximum likelihood estimation (MLE) and visualized using the R language [14]. the results were visualized in R. Linkage disequilibrium (LD) decay was evaluated using PopLDdecay software, which calculated the r2 values of SNPs at various distances, facilitating observation of LD decay with increasing locus separation [15]. Genetic diversity analysis encompassed the proportion of polymorphic markers (PN), expected heterozygosity (HE), observed heterozygosity (HO), and nucleotide diversity (Pi). Heterozygosity and PN analyses were carried out with Plink (v1.9) [16], while Pi was analyzed using VCFtools (v0.1.17) [17].

## Detection of selective features

Based on litter records, the experimental animals were divided into a high-yield group (HL, n = 15) and a low-yield group (LL, n = 15). Selective trait detection was then conducted on these two groups. We employed two methods: the genetic differentiation index ($F_{ST}$) and nucleotide diversity ($\pi$). $F_{ST}$ & $\theta\pi$ analyses were performed using VCFtools (v0.1.17) [18]. Calculations were carried out with a 100kb sliding window and a 10kb step size. First, $F_{ST}$ was calculated, followed by the calculation of $\pi$ values for each group and the $\pi$ ratio ($\pi$ Ratio) between the two groups. Loci present in the top 1% of both the $F_{ST}$ sliding window analysis and the $\theta\pi$ Ratio analysis were identified as candidate target loci.

### Functional annotation and pathway enrichment of candidate genes

After identifying SNP loci in the top 1% for $F_{ST}$ values and π ratios, we obtained the reference genome information for the species from the ENSEMBL website [19]. Using ANNOVAR, we annotated the candidate SNPs to their respective genes. Data filtering excluded loci lacking gene enrichment and those with negative $F_{ST}$ values. Subsequently, annotated genes underwent gene function enrichment analysis using the GO and KEGG databases [20].

## Result

### Population structure and genetic diversity

A genetic analysis comparing Guizhou Black goats and Meigu goats was conducted by using principal component analysis, phylogenetic tree construction, and ancestry components (Fig 1). Principal components 1, 2, and 3 explained 16.63%, 11.27%, and 10.68% of the overall genetic data, respectively. The results of principal component analysis (Figs 1 and 2) and phylogenetic tree (Fig 1) showed that although there was a certain tendency of separation between Qiannan black goats and Meigu goats, there was no significant separation between the groups, as some degree of overlap still persisted. The Guizhou Black goats predominantly clustered together but remained relatively close to the Meigu goats. ADMIXTURE analysis of population structure (Fig 1) demonstrated that the optimal number of clusters is 2. At K = 2, both Meigu goats and Guizhou Black goats displayed two ancestral components, sharing one component. At K = 3, Meigu goats maintained two ancestral components, whereas Guizhou Black goats exhibited three components, sharing two of them. At K = 4, Meigu goats continued with two ancestral components, while Guizhou Black goats showed three components, sharing only one. These findings suggest that Meigu goats generally exhibit lower genetic diversity and more consistent ancestral components. In contrast, Guizhou Black goats likely underwent more migration and admixture events, resulting in the preservation of multiple ancestral components in their genetic background.

The genetic diversity analysis (Fig 3) showed that the polymorphic marker frequency (Pn) was slightly higher in Meigu goats compared to Guizhou black goats. There were no significant differences observed in expected heterozygosity (He), observed heterozygosity (Ho), and nucleotide diversity (Pi) between the two breeds. In terms of linkage disequilibrium (LD) analysis results (Fig 3), the curves for both populations overlap closely, although Guizhou Black goats exhibit slightly higher LD levels compared to Meigu goats.

### Positive selection of high-yield population (HL)

After mutations detection and annotation, 246,490 SNP loci were identified. $F_{ST}$ & θπ analysis (HL vs LL) of these SNPs revealed 33,563 significant loci (Fig 4). Subsequent removal of outliers led to the annotation of 125 candidate genes. Gene Ontology (GO) and Kyoto Encyclopedia of Genes and Genomes (KEGG) pathway analyses identified 94 significantly enriched GO terms and 17 significant (P-value < 0.05) KEGG pathways. The GO terms and KEGG pathways with the lowest P-values were chosen for visualization: the top 10 terms for each GO category (BP, CC, and MF) (Fig 4). The most notable GO term identified was "one-carbon metabolic process" (GO:0006730), associated with diverse physiological functions and metabolic processes. Other GO terms were primarily linked to enzyme activity, metabolic processes, cellular organization, development, and lipid regulation. In the KEGG analysis, the most significant pathway was "One carbon pool by folate" (chx00670), which involves the intracellular transport and utilization of folate. Additional pathways encompassed metabolic processes, signaling pathways, and cellular development.

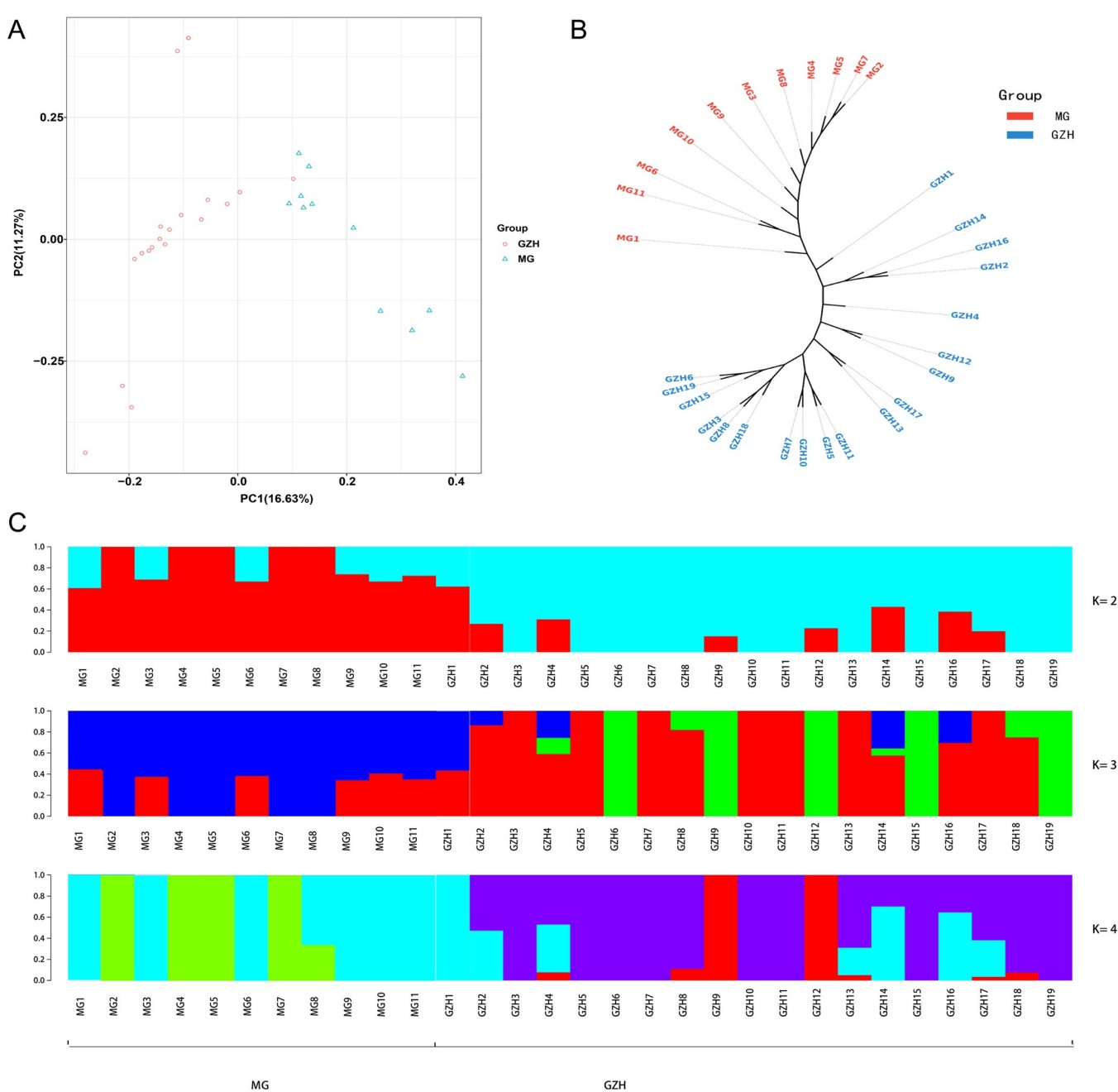

**Fig 1. Population structure and genetic relationships of Guizhou Black goat and Meigu goat.** (A) Principal component analysis and (B) phylogenetic tree were used to classify 30 black goats into two groups. C utilized Admixture software for the analysis of ancestral components, and the optimal K value was 2.

## Candidate genes associated with high litter size

By simultaneously considering both high $F_{ST}$ and high Pi_ratio, candidate genes related to litter performance were identified. These loci showed significant genetic differentiation between groups and maintained high diversity within the high-yield group, indicating their potential importance in reproductive performance. Further, we filtered the $F_{ST}$ and θπ analysis results by removing loci with negative $F_{ST}$ values or other anomalies. We then selected the top 1% of $F_{ST}$ values and SNP loci with Pi_Ratio > 2. Using ANNOVAR for annotation and excluding

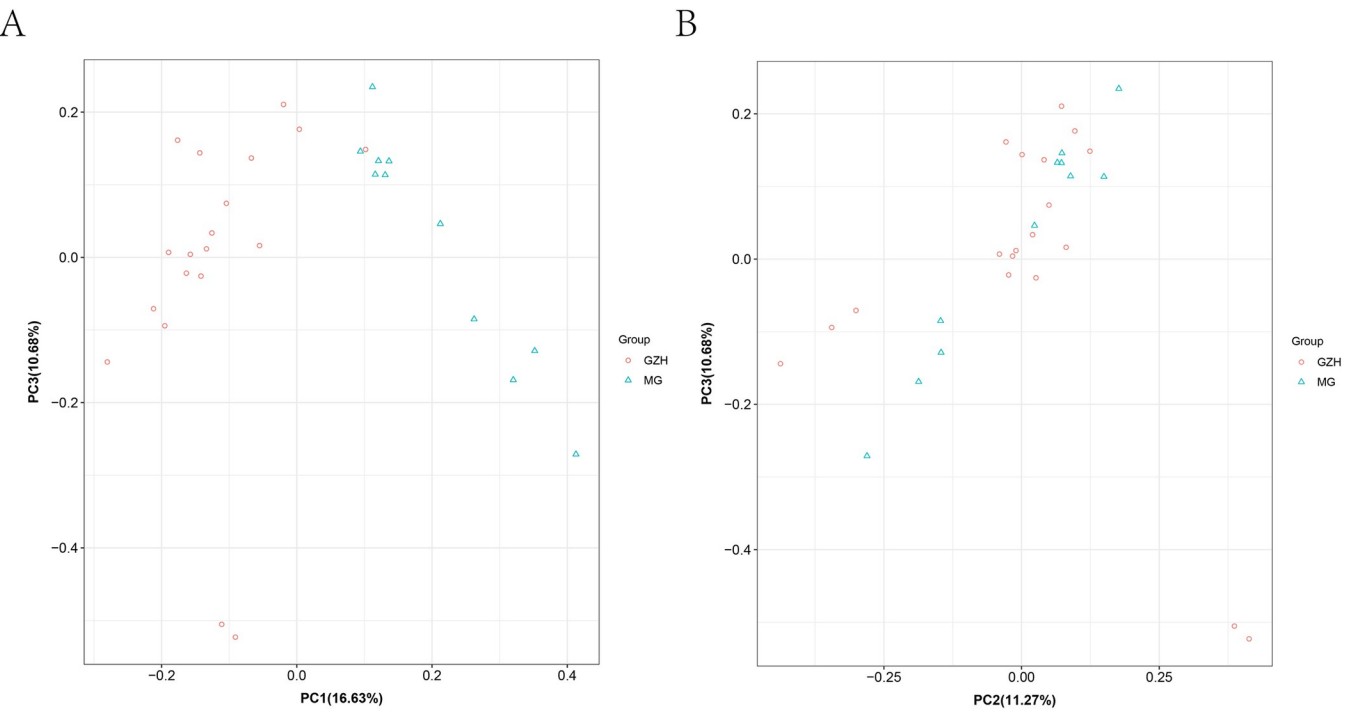

**Fig 2. PCA Supplementary Figure to Fig 1A.**

loci not enriched in genes, we identified six candidate genes: *KCNIP4*, *GFRA2*, *DGKH*, *ARF-GAP3*, *ALDH1L2*, *BFSP2*, and *RNF180*, which may be significantly associated with high litter performance.

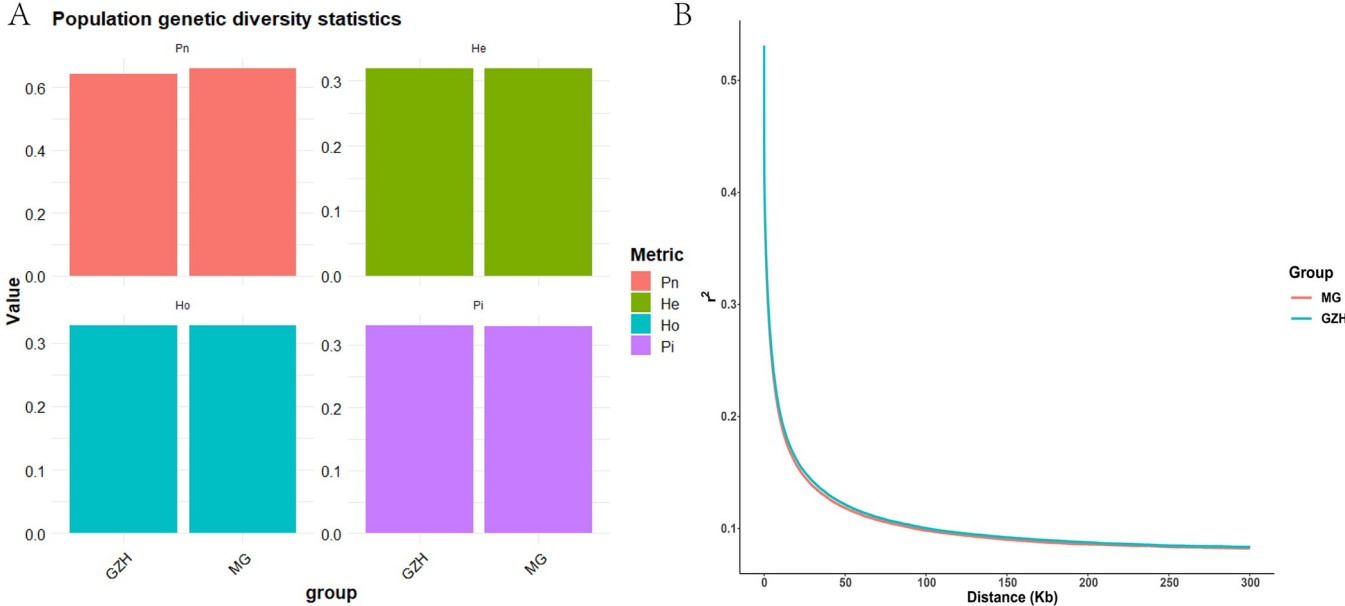

**Fig 3. Genetic diversity analysis and average LD recession.** (A) Proportion of polymorphic markers, expected heterozygosity, observed heterozygosity, and nucleotide diversity of the two goat breeds. (B) Genome-wide average LD recession of the two goat breeds.

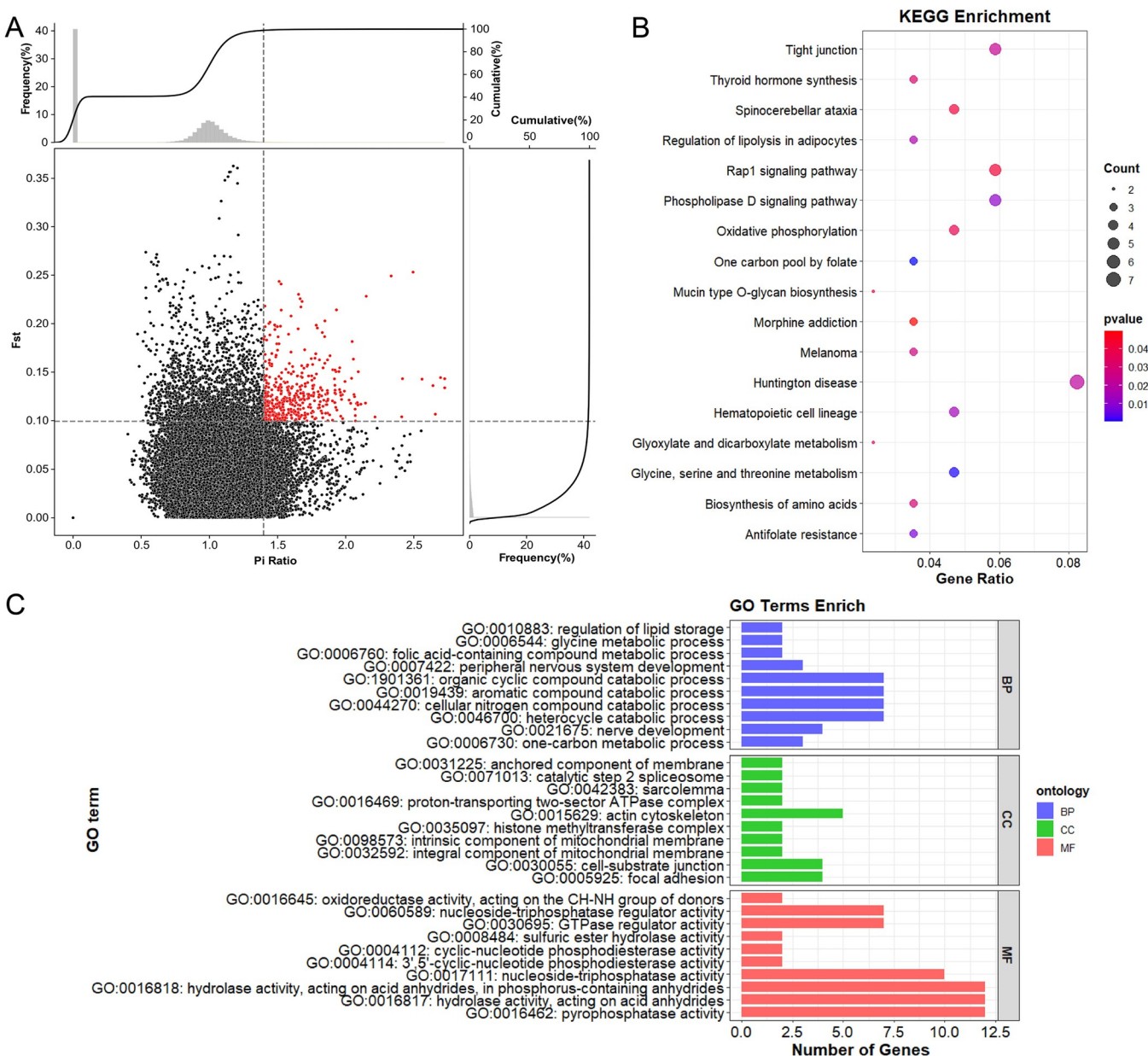

**Fig 4. Positive selection analysis (HL VS LL).** (A) Combining the top 1% Fst values and the top 1% pi ratios to determine the genomic Manhattan selection signature map and the global selection signal map. (B) plots for KEGG pathway analysis. (C) Enriched GO terms BP, CC, MF triple histograms.

## Discuss

Traditional selective breeding in goats is inefficient, time-consuming, and challenging to yield beneficial results in a short term. Therefore, utilizing whole-genome high-throughput sequencing technology can assist in selective breeding and overcome these limitations. Through population structure and genetic diversity analyses, we can understand the evolutionary history and genetic relationships between populations. Selection signal analysis based on litter performance helps identify genes associated with reproduction traits [21,22]. These results can be instrumental in enhancing breeding programs for goats, particularly in addressing the low reproductive performance observed in some indigenous breeds [23].

## Population structure and genetic diversity

The results of principal component analysis showed that some individuals of Guizhou black goat and Meigu goat formed independent clusters, indicating a trend of genetic differentiation between these two breeds. In addition, some individuals of Guizhou black goat were found clustering with Meigu goat in the principal component analysis plot. This indicates the possibility of gene flow or a mixed origin between these two breeds, resulting in a high genetic similarity between the two populations. The separation of Guizhou black goat and Meigu goat into two distinct populations on the phylogenetic tree suggested that these breeds have experienced significant divergence events during their evolution. This indicates that they may have different ancestral origins or have evolved in different geographic or ecological environments. ADMIXTURE analysis confirms a shared ancestry between the breeds, with Guizhou Black goats exhibiting more complex ancestral components compared to Meigu goats. This complexity may be due to long-term hybrid breeding with other high-fertility breeds, leading to increased genetic diversity [4]. Based on the PCA and ADMIXTURE analyses, the genetic diversity indices further suggested that the overall levels of genetic variation were similar in the two breeds, which could be attributed to the fact that the two breeds have maintained a relatively balanced genetic diversity during different evolutionary processes. Based on the above discussion, we conclude that although Guizhou black goats and Meigu goats were once separated in their evolutionary history, they show similarity in genetic structure due to gene flow, mixing, and similar selective pressures in later stages. This complex genetic history and modern environmental influences have jointly contributed to the current genetic structure of these two populations. These findings provided a foundation for further research into the genetic background and selection signals of the two breeds and enhance understanding of their potential genetic mechanisms related to reproductive performance.

## Positive selection of high-yield group and candidate genes for high litter size

Whole-genome selection signal analysis has identified candidate genes associated with high litter performance in Guizhou Black goats and Meigu goats. Enrichment analyses of Gene Ontology (GO) terms and Kyoto Encyclopedia of Genes and Genomes (KEGG) pathways revealed that these enriched terms and pathways are predominantly related to reproductive processes. The most significant GO term identified is "one-carbon metabolic process" (GO:0006730), which is crucial for DNA synthesis and repair, gene methylation, and amino acid metabolism. These processes are essential for embryo health, development, and survival [24,25]. Other important GO terms related to reproduction include "Peripheral nervous system development" (GO:0007422), which regulates the endocrine system and influences hormone secretion, including luteinizing hormone and follicle-stimulating hormone. These hormones directly affect reproductive cycles and litter rates [26]. The most significant KEGG pathway identified is "One carbon pool by folate" (chx00670). The GO enrichment analysis also highlighted "Folic acid-containing compound metabolic process" (GO:0006760), which, like the KEGG pathway, involves critical functions related to folate and its metabolic pathways. Folate deficiency can result in abnormal embryo development, miscarriage, and reduced fertility, thus affecting litter performance [27]. The majority of significant pathways identified are related to reproduction, underscoring the effectiveness of the selection signal analysis in understanding litter performance in goats.

We reviewed relevant literature to analyze the identified candidate genes, finding that most of the six genes are related to animal reproductive performance. Studies have shown that

*KCNIP4* is involved in calcium signaling during the specialization of neural and kidney precursor cells in embryonic development [28]. In the reproductive system, calcium signaling pathways are crucial for ovarian function, oocyte maturation, and embryo development. *KCNIP4* may indirectly influence these processes by regulating calcium signaling, thereby affecting litter rates. *GFRA2*, a specific marker gene for spermatogonial stem cells (SSCs), plays a significant role in testis development and spermatogenesis in cattle [29]. Additionally, increased *GFRA2* expression has been observed in the marginal zone (MZ) of late-stage pregnant mice, indicating its involvement in the recruitment of adult stem cells (SCs) in the MZ, which subsequently leads to the generation of progenitor cell populations [30]. The activity of stem and progenitor cells is crucial for embryonic development and maternal adaptive changes, suggesting that *GFRA2* is closely related to animal reproduction. Researchers, based on experimental data and supporting literature, found that *DGKH* not only regulates the Ras/Raf/MEK/ERK signaling cascade, which controls numerous growth regulatory processes in cells, but also functions downstream in the GnRH signaling pathway [31]. Additionally, transcriptome analysis of germinal vesicle (GV) cattle oocytes post-vitrification revealed significant *DGKH* expression, indicating its role in oocyte growth [32]. *DGKH* may influence litter rates by regulating GnRH signaling and controlling oocyte growth and development. Studies have demonstrated that *ARFGAP3* shows differential expression in the granulosa cells of ovulatory versus dominant follicles in cattle. Specifically, the mRNA expression level of *ARFGAP3* decreases threefold in ovulatory follicles compared to dominant follicles. Although the exact mechanism by which dominant follicles transition to ovulatory follicles is not yet understood, *ARFGAP3* is clearly associated with follicle growth [33]. The number and quality of follicles directly influence litter rates in goats. The oviduct plays a crucial role in litter rates by affecting oocyte transport, fertilization, and early embryo development. *ALDH1L2* displays differential expression in the epithelial cells of the pig oviduct at various stages, with significant upregulation [34], suggesting its involvement in the growth and development of oviduct epithelial cells. The *BFSP2* gene is currently known for its association with lens structure and function, particularly in relation to cataracts. Although direct evidence linking *BFSP2* to reproduction is lacking, it has been implicated in eggshell spot formation [35]. This suggests that its functions extend beyond the lens and cataracts, necessitating further research into its other roles. *RNF180* exhibits significant differential expression between early and late cumulus cells in mice, with upregulation observed in early cumulus cells [36]. This indicates that *RNF180* is crucial in the early stages of follicle development, potentially promoting follicle growth and maturation by regulating cumulus cell function. These studies indicate that most identified candidate genes in the present study are associated with reproduction, directly or indirectly affecting litter traits. While their specific roles in the regulatory network of goat litter rates remain unclear, further research could elucidate their mechanisms influencing reproductive performance.

It is worth noting that fertility is not exclusively related to genetic factors. The effects of environmental factors such as nutrition, environment and management must be taken into account when selecting for high fertility populations. For example, increasing nutritional levels before and during mating has a positive effect on ovulation rates, reproductive losses and lambing rates in goat [37]. Therefore, in this study, the nutritional level, living environment and management practices of the sample goats were controlled to minimize the influence of confounding factors on the experimental results. Meanwhile, epigenetic factors do not require changes in DNA sequence and have a profound effect on fertility by regulating the expression of genes associated with reproduction. Therefore, they should also be taken into account when developing breeding programs.

## Conclusion

We conducted whole-genome high-throughput sequencing (33.3×) on Guizhou Black goats and Meigu goats, identifying a total of 13,275,130 SNPs. Our analysis of population structure and genetic diversity revealed significant genetic differentiation between the two breeds, while their levels of genetic diversity were comparable. Notably, Guizhou Black goats appear to have undergone more extensive admixture and selection events. The analysis of selection signal related to litter performance has identified pathways associated with litter size and highlighted six candidate genes, including *KCNIP4*, *GFRA2*, and *DGKH*, which may be closely linked to enhanced litter performance.

## Supporting information

**S1 Table. 33,563 loci from $F_{ST}$ & θπ analysis.**
(XLSX)

**S2 Table. 125 candidate genes obtained by $F_{ST}$ & θπ analysis.**
(XLSX)

**S3 Table. Significant pathways enriched by KEGG and GO.**
(XLSX)

**S4 Table. SNPs screened for selection signals.**
(XLSX)

**S5 Table. Gene annotation results.**
(XLSX)

**S6 Table. Gene mutation annotation results.**
(ZIP)

**S7 Table. Litter record.**
(XLSX)

**S8 Table. PCA date.**
(XLSX)

## Acknowledgments

Thanks to everyone who helped with this article.

## Author Contributions

**Conceptualization:** Yang Yang.

**Data curation:** Yang Yang.

**Formal analysis:** Chao Yuan.

**Investigation:** Chao Yuan.

**Methodology:** Yanpin Zhao.

**Project administration:** Wen Xiao.

**Software:** Yong Long.

**Supervision:** Yong Han.

**Visualization:** Yanpin Zhao.

**Writing – original draft:** Yanpin Zhao.

**Writing – review & editing:** Yong Han.

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
