## [Decision Letter · Decision Letter 0]

15 Aug 2024

PONE-D-24-30927Genetic Characteristic Analysis and Characteristic Selection of Litter Size in Guizhou Black Goat and Meigu GoatPLOS ONE

Dear Dr. Han,

Thank you for submitting your manuscript to PLOS ONE. After careful consideration, we feel that it has merit but does not fully meet PLOS ONE’s publication criteria as it currently stands. Therefore, we invite you to submit a revised version of the manuscript that addresses the points raised during the review process.

Please submit your revised manuscript by Sep 29 2024 11:59PM. If you will need more time than this to complete your revisions, please reply to this message or contact the journal office at plosone@plos.org. Please include the following items when submitting your revised manuscript:A rebuttal letter that responds to each point raised by the academic editor and reviewer(s). You should upload this letter as a separate file labeled 'Response to Reviewers'.A marked-up copy of your manuscript that highlights changes made to the original version. You should upload this as a separate file labeled 'Revised Manuscript with Track Changes'.An unmarked version of your revised paper without tracked changes. You should upload this as a separate file labeled 'Manuscript'.

We look forward to receiving your revised manuscript.

Kind regards,

Xiaona Wang, Ph.D

Academic Editor

PLOS ONE

Journal Requirements:

The name of the colleague or the details of the professional service that edited your manuscript.A copy of your manuscript showing your changes by either highlighting them or using track changes (uploaded as a *supporting information* file).A clean copy of the edited manuscript (uploaded as the new *manuscript* file).

"1.Innovative Utilization of Livestock and Poultry Germplasm Resources and Construction of Basic Science and Technology Platform (Qiannongke Technology Innovation [2023] 04)

2.Major Special Projects of the Guizhou Province Department of Science and Technology provided funding for this work (Qianke Service Enterprise [2020] 4009)."

6. Please upload a copy of Figures 1 (a-c), 2, and 3 (a-c) to which you refer in your text on pages 17 and 18 (in PDF format). If the figure is no longer to be included as part of the submission please remove all reference to it within the text.

Reviewer's Responses to Questions

**Comments to the Author**

1. Is the manuscript technically sound, and do the data support the conclusions?

Reviewer #1: Yes

Reviewer #2: Yes

Reviewer #3: Yes

Reviewer #4: Partly

Reviewer #5: Yes

2. Has the statistical analysis been performed appropriately and rigorously? 

Reviewer #1: Yes

Reviewer #2: Yes

Reviewer #3: N/A

Reviewer #4: N/A

Reviewer #5: Yes

3. Have the authors made all data underlying the findings in their manuscript fully available?

Reviewer #1: Yes

Reviewer #2: No

Reviewer #3: Yes

Reviewer #4: No

Reviewer #5: Yes

4. Is the manuscript presented in an intelligible fashion and written in standard English?

Reviewer #1: Yes

Reviewer #2: No

Reviewer #3: Yes

Reviewer #4: Yes

Reviewer #5: Yes

5. Review Comments to the Author

Reviewer #1: Issues in the research: It is advisable to clarify the gender of goats. Should the default gender assumption be female? Could genetic variations in rams impact litter size when a goat and its offspring are involved? The study lacks rigor as it does not fully consider the influence of male populations in mammalian breeding. The manuscript title could be refined for clarity. In the discussion section, it is recommended to focus on the same species. Many of the cited sources refer to species other than goats. Quoting conclusions from such studies may mislead readers and provide inaccurate guidance.

Reviewer #2: 1. A fundamental issue that cannot be ignored is the variability in kidding records between the 11meigu goats and 19 Guizhou Black goats. The author has divided them into two groups based on kidding records, suggesting that the kidding rates of these two goat breeds themselves are unstable. Why did the study use two different breeds for the same research and not group them by breed? The inherent differences between these breeds and individuals are difficult to eliminate in sequencing results. The author is urged to explain in detail the rationale and feasibility behind this approach.Especially notable is the distinct clustering observed between Guizhou Black goats and Meigu goats in the PCA analysis, suggesting substantial genetic differentiation between these two breeds.

2.Please provide a data availability statement; evidently, the authors have not provided accessible pathways to the original sequencing data.

3.The numbering of section headings throughout the manuscript is incorrect.

4.Figures and tables that should appear in the main text should not be included as supplementary materials.

5.Regarding the results of GO and KEGG analyses, descriptions of enriched functions and pathways should be based on grouped traits. However, these descriptions should remain objective, avoiding subjective interests of the authors or traits relevant only to the objective.

6.The results in Tables 1 and 2 are unacceptable, lacking any valid informational basis beyond the names of genes. Please provide detailed numerical values to ensure the credibility of the results.

Reviewer #3: Zhao et al. analyzed the genetic structure of guizhou black goat and meigu goat, and selection signal of their litter size. The results showed that the two population have similar level of genetic diversity. Meanwhile, six candidate genes were identified through selection signal analysis. However, some questions should be addressed.

Detailed comments:

Line 39-52, Guizhou black goat and Meigu goat have been used to research material, but we don’t know their genetic background and why they have been selected. Please explain it.

Line 97-100, because there is significant differentiation between guizhou black goat and meigu goat, it is unreasonable that the two populations were categorized into high-yield and low-yield groups.

Line 251-255, the two breeds all have small samples (11 vs 19) and different sample size, so the number of samples should be discussed.

In Figure 2(A), the significant statistical analysis should be needed.

Reviewer #4: This study was collected 19 Guizhou Black goats and 11 Meigu goats for whole genome high-throughput sequencing, and to explore their genetic characteristics, structure through population genetics analysis, and to reproductive performance through selection signal analysis.The results identified six candidate genes significantly associated with high litter performance, like KCNIP4, GFRA2, and DGKH. However, this manuscript is not yet sufficient for publication.

1.This manuscript collected two populations of Guizhou Black goats and Meigu goats, but it lacks a description of the breed characteristics and genetic background of these two populations, such like the Meigu goat is native to Meigu County, Sichuan Province.

2.In the part of population genetics analysis, the results of population structure show that Guizhou Black goats exhibiting more complex ancestral components compared to Meigu goats. Actually, that is not suitable for the signal analysis of high and low reproductive group selection. Suggestion 1: Add an out-group with a distant genetic background for analysis, such as Boer goats 2: Add kinship matrix analysis.

3.This manuscript conducted selection signal analysis on high and low reproductive groups, but it lacks details grouping information and description of the standards for high and low reproduction, also due to the different genetic backgrounds of the two populations. So, it is difficult to obtain reliable analysis results. Firstly,the author can supplement a certain sample size, with low depth sequencing,to 30 samples each population.secondly, it is suggest conducting selection signal analysis within the population,then take the intersection of the results of the two population.

4.This manuscript was used more in-depth sequencing, and it is recommended to supplement comparative analysis of structural variations, especially CNV.

5.This manuscript was identified 246,490 SNP loci, and revealed 33,563 significant loci. Curiously,there were only annotation 125 candidate genes. Please describe the annotation process and provide a list of 125 genes.

6.The supporting information Table 1,2 and 3.csv were not occured in this manuscript.Please confirm that.

Reviewer #5: The manuscript presents a comprehensive study on the genetic characteristics of Guizhou Black goats and Meigu goats, focusing on their reproductive performance. This can provide insights into the understanding of the genetic basis of reproductive performance in the two goat breeds. However, the manuscript could be strengthened and improved by addressing the limitations as below.

Major issues:

1. The "Materials and Methods" section does not specify the age, sex, and health status of the goats. It is also unclear how representative the sampled individuals are of the broader populations of Guizhou Black and Meigu goats.

2. The "Discussion" section assumes a direct causal relationship between identified genetic markers and litter performance. However, correlation does not imply causation, and the possibility of confounding factors or linkage disequilibrium affecting the results is not thoroughly discussed.

3. The "Discussion" section focuses primarily on genetic factors. The influence of environmental factors, such as nutrition, climate, and management practices, on reproductive performance is not considered. Epigenetic factors, which can also play a significant role in reproductive traits, are not discussed.

4. The "Discussion" section identifies candidate genes but does not provide evidence of their functional validation in the context of goat reproduction.

Minor issues:

5. The method of DNA extraction is mentioned but not detailed enough. Specific DNA extraction kits and protocols should be mentioned.

6. The use of DNBSEQ technology by BGI is mentioned, but the model and version of the sequencer should be specified.

7. The criteria for high-quality loci are mentioned (detection rate ≥ 90%, minor allele frequency ≥ 0.05), but the rationale for these thresholds is not explained.

8. The Admixture software is used for population structure analysis, but the algorithm and the number of clusters used are not specified.

9. The criteria for defining high-yield and low-yield groups based on litter records are not detailed.

10. The methods used for FST and π calculations are mentioned, but the statistical tests used to analyze the results are not described.

6. PLOS authors have the option to publish the peer review history of their article (what does this mean?). If published, this will include your full peer review and any attached files.

Reviewer #1: No

Reviewer #2: No

Reviewer #3: No

Reviewer #4: No

Reviewer #5: No

---

## [Author Response · Author response to Decision Letter 0]

28 Aug 2024

List of Responses

Dear editors and reviewers:

Thank you very much for your careful review and constructive suggestions with regard to our manuscript “Genetic Characteristic Analysis and Characteristic Selection of Litter Size in Guizhou Black Goat and Meigu Goat (PONE-D-24-30927)”. We have carefully evaluated the Editors/Reviewers’ critical comments and thoughtful suggestions, responded to these suggestions point-by-point, and revised the manuscript accordingly. The revised parts in the manuscript have been marked in red. We appreciate for Editors/Reviewers’ warm work earnestly, and hope that the corrections will meet with approval. Please feel free to contact us with any questions and we are looking forward to your consideration. The main corrections in the manuscript and the responses to the Editors/Reviewers’ comments are as follows.

Response：The manuscript and supporting materials were revised with reference to the PLos One submission guidelines.

2. We suggest you thoroughly copyedit your manuscript for language usage, spelling, and grammar.

Response：The manuscript was thoroughly proofread for language usage, spelling and grammar. Providing embellishment services was Yi Fu Zhao, Guizhou Institute of Animal Husbandry and Veterinary Medicine, Guizhou, China.

3. In your Methods section, please provide additional information regarding the permits you obtained for the work.

Response：It was confirmed with the relevant institutions that common venous blood collection, no duplicate blood collection, no large amount of blood collection (only 3 ml), and no selection of blood collection methods such as funduscopic blood collection, which may have an impact on the health and welfare of experimental animals, do not require experimental animal welfare and ethical review.

4. Please state what role the funders took in the study.

Response：Already added financial disclosure statement to cover letter.

5. PLOS requires an ORCID iD for the corresponding author in Editorial Manager on papers submitted after December 6th, 2016. Please ensure that you have an ORCID iD and that it is validated in Editorial Manager.

Response：Already certified as required

6. Please upload a copy of Figures 1 (a-c), 2, and 3 (a-c) to which you refer in your text on pages 17 and 18 (in PDF format).

Response：The image (PDF) has been uploaded and some quotes have been removed from the article.

Reviewer #1:

 Issues in the research: It is advisable to clarify the gender of goats. Should the default gender assumption be female? Could genetic variations in rams impact litter size when a goat and its offspring are involved? The study lacks rigor as it does not fully consider the influence of male populations in mammalian breeding. The manuscript title could be refined for clarity. In the discussion section, it is recommended to focus on the same species. Many of the cited sources refer to species other than goats. Quoting conclusions from such studies may mislead readers and provide inaccurate guidance.

Response：1. The goats used in this study were all females and have been supplemented in material methods. 2. Males were not collected because it was not possible to collect litter sizes to determine whether they were more or less abundant. 3. The title has been changed for greater clarity. 4. I do my best to minimize references to species other than goats in my discussions, but there are cases where there is no research on goats.

Reviewer #2

1. A fundamental issue that cannot be ignored is the variability in kidding records between the 11meigu goats and 19 Guizhou Black goats. The author has divided them into two groups based on kidding records, suggesting that the kidding rates of these two goat breeds themselves are unstable. Why did the study use two different breeds for the same research and not group them by breed? The inherent differences between these breeds and individuals are difficult to eliminate in sequencing results. The author is urged to explain in detail the rationale and feasibility behind this approach.Especially notable is the distinct clustering observed between Guizhou Black goats and Meigu goats in the PCA analysis, suggesting substantial genetic differentiation between these two breeds.

Response：These two breeds were used because their habitat and phenotypic characteristics are very similar, except for a large difference in fecundity, and because both breeds have a similar genetic structure. I apologize for not including all the results of the principal component analysis in the article in order to keep the structure of the article concise. In fact, the results of the principal component analysis showed no significant differences between the two species, and our ANOVA test following the PCA showed that the two populations were significantly different only in principal component 1, but not in principal components 2 and 3. I have uploaded the PCA data into the supporting materials (Table 8).

2.Please provide a data availability statement; evidently, the authors have not provided accessible pathways to the original sequencing data.

Response：I apologize for the problem with the access path to the raw sequencing data, I need to get permission from the relevant authorities to upload the raw data. The only currently available route is to contact me directly at 2642900225@qq.com.

3.The numbering of section headings throughout the manuscript is incorrect.

Response：Corrections were made to the numbering of section headings throughout the manuscript

4.Figures and tables that should appear in the main text should not be included as supplementary materials.

Response：Figures and tables appearing in the main text have been removed from the supplementary material

5.Regarding the results of GO and KEGG analyses, descriptions of enriched functions and pathways should be based on grouped traits. However, these descriptions should remain objective, avoiding subjective interests of the authors or traits relevant only to the objective.

Response：The descriptions of GO and KEGG analysis results have been modified accordingly to make them more objective.

6.The results in Tables 1 and 2 are unacceptable, lacking any valid informational basis beyond the names of genes. Please provide detailed numerical values to ensure the credibility of the results.

Response：Allele information has been added in Table 1 and supplemented with SNPs screened for selection signals (Table4) and gene annotation results (Table5).

Reviewer #3 

1. Guizhou black goat and Meigu goat have been used to research material, but we don’t know their genetic background and why they have been selected. Please explain it.

Response：Genetic background and reasons for selection added to the introduction

2. because there is significant differentiation between guizhou black goat and meigu goat, it is unreasonable that the two populations were categorized into high-yield and low-yield groups.

Response：I apologize for not including all the results of the principal component analysis in the article in order to keep the structure of the article concise. In fact, the results of the principal component analysis showed no significant differences between the two species, and our ANOVA test following the PCA showed that the two populations were significantly different only in principal component 1, but not in principal components 2 and 3. I have uploaded the PCA data into the supporting materials (Table 8).

3. the two breeds all have small samples (11 vs 19) and different sample size, so the number of samples should be discussed.

Response：Due to financial constraints, the number of samples was small. However, we found that a small number of samples could be used for population structure analysis, and the corresponding references are as follows Zhang T, Wang Z, Li Y, et al. Genetic diversity and population structure of five populations of cashmere goats in Inner Mongolia using genome-wide genotyping. Mongolia using genome-wide genotyping. Anim Biosci. 2024; 37(7):1168-1176. doi: 10.5713/ab.23.0424. 10.5713/ab.23.0424. Population structure analysis and genetic diversity analysis can be performed even if the number of samples varies.

4.In Figure 2(A), the significant statistical analysis should be needed.

Response：The genetic diversity analyses in Fig. 2(A) are based on high- and low-yielding populations, not individuals, and therefore could not be analyzed statistically for significance.

Reviewer #4

1.This manuscript collected two populations of Guizhou Black goats and Meigu goats, but it lacks a description of the breed characteristics and genetic background of these two populations, such like the Meigu goat is native to Meigu County, Sichuan Province.

Response：A description of the breed characteristics and genetic background of the black goat and the Meigu goat in Guizhou has been added to the introduction.

2.In the part of population genetics analysis, the results of population structure show that Guizhou Black goats exhibiting more complex ancestral components compared to Meigu goats. Actually, that is not suitable for the signal analysis of high and low reproductive group selection. 

Suggestion 1: Add an out-group with a distant genetic background for analysis, such as Boer goats. Suggestion 2: Add kinship matrix analysis. Suggestion 3.This manuscript conducted selection signal analysis on high and low reproductive groups, but it lacks details grouping information and description of the standards for high and low reproduction, also due to the different genetic backgrounds of the two populations. So, it is difficult to obtain reliable analysis results. Firstly,the author can supplement a certain sample size, with low depth sequencing,to 30 samples each population.secondly, it is suggest conducting selection signal analysis within the population,then take the intersection of the results of the two population.

Response：Thank you very much for your thorough review of our manuscript and your valuable suggestions. We have carefully considered your suggestions and recognize that they are very valuable. However, due to limited funding and analytical equipment, we were unable to supplement the study accordingly. Nevertheless, your suggestions will be a very important reference for us to conduct similar studies in the future.

Suggestion 4.This manuscript was used more in-depth sequencing, and it is recommended to supplement comparative analysis of structural variations, especially CNV.

Response：The results of the CNV comparative analysis have been used by other colleagues in the subject group and therefore cannot be added.

Suggestion 5.This manuscript was identified 246,490 SNP loci, and revealed 33,563 significant loci. Curiously,there were only annotation 125 candidate genes. Please describe the annotation process and provide a list of 125 genes.

Response：For details on the annotation process, see the “Functional Annotation and Pathway Enrichment of Candidate Genes” content, and for a list of the 125 candidate genes, see Table 3 of the Supporting Materials.

Suggestion 6.The supporting information Table 1,2 and 3.csv were not occured in this manuscript.Please confirm that.

Response：I have submitted the support material in the system, it may be a display problem, please try to get it again.

Reviewer #5

1. The "Materials and Methods" section does not specify the age, sex, and health status of the goats. It is also unclear how representative the sampled individuals are of the broader populations of Guizhou Black and Meigu goats.

Response：Age, sex and health status of the goats were also added. The samples of Guizhou black goats came from the government experimental breeding farm; the Meigu goats, although from Guizhou, were introduced from Meigu County, Sichuan, and both are representative.

2. The "Discussion" section assumes a direct causal relationship between identified genetic markers and litter performance. However, correlation does not imply causation, and the possibility of confounding factors or linkage disequilibrium affecting the results is not thoroughly discussed.

Response：The content of the “Discussion” has been revised accordingly to make its description more objective.

3. The "Discussion" section focuses primarily on genetic factors. The influence of environmental factors, such as nutrition, climate, and management practices, on reproductive performance is not considered. Epigenetic factors, which can also play a significant role in reproductive traits, are not discussed.

Response：Every effort was made to maintain consistency in the rearing environment, nutritional levels, and management practices of the samples collected, and these factors were not addressed in the discussion. The effects of the corresponding factors on goat fertility have been added to the discussion.

4. The "Discussion" section identifies candidate genes but does not provide evidence of their functional validation in the context of goat reproduction.

Response：This is because there is no literature on validation studies of the screened candidate genes for reproductive function in goats.

Minor issues:

5. The method of DNA extraction is mentioned but not detailed enough. Specific DNA extraction kits and protocols should be mentioned.

Response：Some non-essential content has been omitted to make the article more concise. The DNA extraction kit used for the magnetic bead assay is the QIAGEN DNeasy Blood and Tissue Kit. The procedure is Sample Preparation: Blood samples are mixed well and centrifuged.Lysis: Blood samples are mixed with lysis buffer and protease and incubated at 55°C for 1 hour to allow complete cell lysis. Magnetic Bead Binding: The lysate is added to a reagent containing magnetic beads and the DNA is bound to the beads by a magnetic field. Wash: Wash the beads with wash buffer to remove impurities. Elution: The DNA on the beads is eluted with elution buffer to obtain a high purity DNA solution. Quality Control: Quality control is performed using agarose gel electrophoresis and enzyme labeling assays.

6. The use of DNBSEQ technology by BGI is mentioned, but the model and version of the sequencer should be specified.

Response：the sequencer model is DNBSEQ-T7.

7. The criteria for high-quality loci are mentioned (detection rate ≥ 90%, minor allele frequency ≥ 0.05), but the rationale for these thresholds is not explained.

Response：By setting the quality control conditions of detection rate ≥ 90% and minimum allele frequency ≥ 0.05 in Plink, the quality of genotype data can be effectively improved to ensure the reliability and statistical efficacy of the results of subsequent analysis. These criteria are usually used as quality control measures for genomic association studies, population genetics analysis, etc.

8. The Admixture software is used for population structure analysis, but the algorithm and the number of clusters used are not specified.

Response：Population structure analysis was based on maximum likelihood estimation (MLE) to infer the ancestral components of individuals. The optimal number of clusters was 2.

9. The criteria for defining high-yield and low-yield groups based on litter records are not detailed.

Response：Litter records uploaded to support materials (Table 7)

10. The methods used for FST and π calculations are mentioned, but the statistical tests used to analyze the results are not described.

Response：The results of the Fst and π-value calculations themselves do not need to be directly tested statistically, and can be used directly in selective analyses, requiring only the design of thresholds.

We have tried our best to improve the manuscript and made some changes in the manuscript. We appreciate for Editors/Reviewers’ warm work earnestly, and hope that the correction will meet with approval. Once again, thank you very much for your comments and suggestions.

Yours sincerely

---

## [Decision Letter · Decision Letter 1]

22 Oct 2024

Genetic Characterization and Selection of Litter Size Traits of Guizhou Black Goat and Meigu Goat

PONE-D-24-30927R1

Dear Dr. Han,

The genetic background inconsistency of grouping by lambing number across different breeds did not undermine the conclusions drawn.  We’re pleased to inform you that your manuscript has been judged scientifically suitable for publication and will be formally accepted for publication once it meets all outstanding technical requirements. 

Kind regards,

Xiaona Wang, Ph.D

Academic Editor

PLOS ONE

Additional Comments from the Editor:

1. One reviewer rejected the manuscript while three reviewers recommended publication. The quality of feedback to authors was reasonable. These findings enhanced the understanding of the genetic characteristics and population structure of Guizhou Black, goats and Meigu goats, which possess innovation and significance. Based on these factors, I chose to accept the paper.

2. With regards to genetic background inconsistency when grouping by lambing number across different breeds, the author explains it in detail as follows.

These two breeds were used because their habitat and phenotypic characteristics are very similar, except for a large difference in fecundity, and because both breeds have a similar genetic structure. In fact, the results of the principal component analysis showed no significant differences between the two species, and our ANOVA test following the PCA showed that the two populations were significantly different only in principal component 1, but not in principal components 2 and 3. Authors have uploaded the PCA data into the supporting materials (Table 8).

**Comments to the Author**

1. If the authors have adequately addressed your comments raised in a previous round of review and you feel that this manuscript is now acceptable for publication, you may indicate that here to bypass the “Comments to the Author” section, enter your conflict of interest statement in the “Confidential to Editor” section, and submit your "Accept" recommendation.

Reviewer #1: All comments have been addressed

Reviewer #3: All comments have been addressed

Reviewer #5: All comments have been addressed

2. Is the manuscript technically sound, and do the data support the conclusions?

Reviewer #1: Yes

Reviewer #3: Yes

Reviewer #5: Yes

3. Has the statistical analysis been performed appropriately and rigorously? 

Reviewer #1: Yes

Reviewer #3: Yes

Reviewer #5: Yes

4. Have the authors made all data underlying the findings in their manuscript fully available?

Reviewer #1: Yes

Reviewer #3: Yes

Reviewer #5: Yes

5. Is the manuscript presented in an intelligible fashion and written in standard English?

Reviewer #1: Yes

Reviewer #3: Yes

Reviewer #5: Yes

6. Review Comments to the Author

Reviewer #1: The questions I raised have been revised by the authors. It may be published if requirements for publication are met.

Reviewer #3: I have no questions for this revised manuscript. I agree with the acceptance of this revised manucript.

Reviewer #5: The authors addressed all issues proposed by the reviewers and thus I recommend accept the revised manuscript for publication.

7. PLOS authors have the option to publish the peer review history of their article (what does this mean?). If published, this will include your full peer review and any attached files.

Reviewer #1: No

Reviewer #3: No

Reviewer #5: **Yes: **Shanyuan Chen

---

## [Editor Report · Acceptance letter]

25 Oct 2024

PONE-D-24-30927R1 

PLOS ONE

Dear Dr. Han, 

I'm pleased to inform you that your manuscript has been deemed suitable for publication in PLOS ONE. Congratulations! Your manuscript is now being handed over to our production team.

Kind regards, 

on behalf of

Associate Professor Xiaona Wang 

Academic Editor

PLOS ONE